# Novel, Fluorine-Free Membranes Based on Sulfonated Polyvinyl Alcohol and Poly(ether-block-amide) with Sulfonated Montmorillonite Nanofiller for PEMFC Applications

**DOI:** 10.3390/membranes14100211

**Published:** 2024-10-01

**Authors:** Manhal H. Ibrahim Al-Mashhadani, Gábor Pál Szijjártó, Zoltán Sebestyén, Zoltán Károly, Judith Mihály, András Tompos

**Affiliations:** 1Institute of Materials and Environmental Chemistry, HUN-REN Research Centre for Natural Sciences, Magyar Tudósok Körútja 2, H-1117 Budapest, Hungary; manhal.ibrahim@ilps.uobaghdad.edu.iq (M.H.I.A.-M.); szijjarto.gabor@ttk.hu (G.P.S.); sebestyen.zoltan@ttk.hu (Z.S.); karoly.zoltan@ttk.hu (Z.K.); mihaly.judith@ttk.hu (J.M.); 2Hevesy György Doctoral School of Chemistry, Eötvös Loránd University, Pázmány Péter sétány 1/A, H-1117 Budapest, Hungary; 3Institute of Laser for Postgraduate Studies, University of Baghdad, Baghdad 10070, Iraq

**Keywords:** proton exchange membrane, inorganic–organic hybrid, fuel cell, fluorine free, sulfonated polyvinyl alcohol (S-PVA), PEBAX, sulfonated montmorillonite (S-MMT)

## Abstract

Novel blend membranes containing S-PVA and PEBAX 1657 with a blend ratio of 8:2 (referred to as SPP) were prepared using a solution-casting technique. In the manufacturing process, sulfonated montmorillonite (S-MMT) in ratios of 0%, 3%, 5%, and 7% was used as a filler. The crystallinity of composite membranes has been investigated by X-ray diffraction (XRD), while the interaction between the components was evaluated using Fourier-transform infrared spectroscopy (FT-IR). With increasing filler content, good compatibility between the components due to hydrogen bonds was established, which ultimately resulted in improved tensile strength and chemical stability. In addition, due to the sulfonated moieties of S-MMT, the highest ion exchange capacity (0.46 meq/g) and water uptake (51.61%) can be achieved at the highest filler content with an acceptable swelling degree of 22.65%. The composite membrane with 7% S-MMT appears to be suitable for application in proton exchange membrane fuel cells (PEMFCs). Amongst the membranes studied, this membrane achieved the highest current density and power density in fuel cell tests, which were 149.5 mA/cm^2^ and 49.51 mW/cm^2^. Our fluorine-free composite membranes can become a promising new membrane family in PEMFC applications, offering an alternative to Nafion membranes.

## 1. Introduction

As technology advances and the population increases, the demand for and dependence on energy increases accordingly. Fuel cells offer a promising method for energy conversion that could satisfy this growing demand for energy efficiently [1]. Proton exchange membrane fuel cells (PEMFCs) are becoming increasingly popular due to their advantageous operational conditions, versatility in applications, and notable efficiency [2]. The proton exchange membrane (PEM) is often considered the “heart” of a PEM fuel cell due to its critical function in the cell’s operation [3,4,5]. Membranes to be deemed appropriate for use in PEMFCs are required to demonstrate certain characteristics, such as elevated proton conductivity, low fuel crossover, resilience to thermal and chemical wear, strong mechanical strength, and affordability [6]. Perfluorosulfonic (PFSA) polymers, under the trade name Nafion, are widely utilized as commercial membranes, primarily because of their exceptional mechanical robustness and superior proton conductivity [7,8,9]. Hydrated membranes boost proton mobility due to the presence of solvated sulfonic acid groups and the microphase separation of hydrophilic groups and the hydrophobic polytetrafluoroethylene (PTFE) backbone. Additionally, the presence of a PTFE backbone contributes significantly to mechanical robustness [10]. However, Nafion is expensive and loses water at higher operating temperatures [11], so researchers are looking for suitable alternatives [12]. One approach for improving the performance of the proton exchange membrane is to use a blend of different polymers [13]. PEMFC membranes have been constructed with a variety of hydrocarbon polymers as the primary polymer chain, including polyether ether ketone (PEEK) [14], polyarylene ether sulfone [15], polybenzimidazole (PBI) [16], polyether sulfone (PES) [17], polyamide [18], and polyvinyl alcohol (PVA) [19]. PVA stands out among these polymers because it is highly hydrophilic, biodegradable, easy to modify, and has excellent film-forming properties [20,21,22]; although, it has some drawbacks, such as being easily soluble in water, having a low strength, and having a low thermal stability. As a result, it cannot provide all the needed membrane quality indicators by itself [23,24,25]. PVA modified with sulfonic acid groups has potential proton exchange membrane properties. Accordingly, the addition of a sulfonic acid-containing crosslinking agent, such as sulfosuccinic acid (SSA), can be employed to enhance the proton conductivity and mechanical stability of PVA membranes. SSA content was thoroughly investigated [26,27]. The studies revealed the suitability of these crosslinked polymers for fuel cell applications [28].

Poly(ether-block-amide) polymers sold under the brand name PEBAX^®^ are widely applied because they have beneficial properties such as high permeability, high selectivity, and long-lasting durability [29]. Polymers’ architecture consists of chemically bonded rigid polyamide and flexible polyether segments, creating a thermoplastic copolymer [30]. The mechanical stability of the copolymer is governed by the crystalline configuration of the polyamide segments. Conversely, the amorphous nature of the polyether segments facilitates material transport [31].

PEBAX copolymers are produced by copolymerizing different poly(amides), including PA6, PA12, and PA66, with various poly(ethers), such as polyethylene oxide (PEO) and poly(tetramethylene oxide) [32]. PEBAX 1657 consists of 40% amide and 60% ether segments. The unique crystalline-amorphous framework of PEBAX 1657 combines the features of thermoplastics and elastomers, leading to a composite with enhanced mechanical durability and chemical resistance [33]. To the best of our knowledge, only a few articles have discussed the role of PEBAX as a membrane for fuel cell applications [29,34,35,36,37]. The application of PEBAX has been investigated in the dehydration and humidification of inlet gases, which provide a balance of plants in PEM fuel cells, and also as a proton exchange membrane.

Composites of cerium fluoride oxide two-dimensional (2D) mesoporous nanosheets and 1-ethyl-3-methylimidazolium dicyanamide ionic liquids (IL@F-Ce) were introduced as fillers into PEBAX^®^ 1074. The slit-shaped mesoporous structure of the nanosheets facilitates the construction of water vapor rapid transport channels in blend membranes [37].

Poly(oxyalkylene)amine (APOP) was used to modify montmorillonite (MMT) to improve the compatibility of PEBAX ^®^ 1074 without reducing the hydrophilicity of MMT [35]. Therefore, it is worthwhile to try this kind of composite membrane as a proton exchange membrane in PEMFCs, especially in portable devices when open cathodes should meet the requirements of low humidity.

As a proton exchange membrane, a polyion complex membrane for direct methanol fuel cell application was synthesized by blending the natural biopolymer sodium alginate with PEBAX 1657. Of decisive importance was the development of proton conductivity, which was achieved by sulfonating the blend with sulfuric acid (H_2_SO_4_). The sulfonated blend membrane exhibited a high ion exchange capacity of 2.1 meq/g and a high proton conductivity of 0.067 S/cm [36].

Montmorillonite (MMT) is a stratified silicate composed of silica and alumina octahedra layers. It can be incorporated into polymer nanocomposites. Its vast surface area, impressive two-dimensional nanostructures, and adaptable layered characteristics enable modification of MMT’s layered structure by chemical and physical methods [35]. Before usage in membranes, MMT was generally modified. In PEM, the application of sulfonated silicate can be advantageous. The presence of sulfonic acid and hydroxyl end groups on the surface of MMT contributes to proton conductivity and water uptake and controls the swelling ratio. However, the compatibility of these groups with the polymer matrix can be challenging [38].

The combination of advantageous properties of PEBAX, sulfonated PVA, and sulfonated MMT may lead to proton exchange membranes with practical importance. In the proposed blend membrane, PEBAX 1657, having both hydrophilic and hydrophobic segments, will provide morphological and mechanical stability. Additionally, this component provides channels for the transfer of protons due to the separation of the two phases. In S-PVA, SSA serves as a crosslinker and also provides sulfonic acid groups [39], which makes the membrane more hydrophilic while keeping the swelling level for PVA under control. Moreover, the aim of this work is to study the effect of different amounts of S-MMT on water uptake, swelling degree, thermal stability, chemical stability, ion exchange capacity, and performance of the blended membranes in fuel cells. The new composite membranes can provide an alternative to fluorine-based PEMs and thus help fuel cell technologies progress towards better, more environmentally friendly solutions.

## 2. Experimental Methodology

### 2.1. Materials

Arkema (Colombes, France) provided PEBAX 1657. PEBAX 1657 consists of polyether oxide (PEO) (60 wt%) and polyamide (PA6) (40 wt%). High-molecular-weight PVA (Mw = 85,000–124,000 g/mol) with hydrolyzed 99 + % was obtained from Sigma-Aldrich (USA). The solution of SSA (70% in water) was obtained from Sigma-Aldrich (USA) too, and it was used without any purification. The source of the Na montmorillonite (MMT) was Shandong Yousuo Chemical Co., Ltd., located in Heze City, Shandong Province, China. A 5 wt% copolymer resin solution of DuPont Nafion (D520–1000 EW) was acquired from the fuel cell store. Dimethyl acetamide (DMAc) was sourced from VWR Chemicals, located in Budapest, Hungary. The solvents, absolute ethanol, and 2-propanol were acquired from Gilca, Lab Box, and Schar Lab, located in Barcelona, Spain, respectively. Millipore water was used to prepare all the aqueous solutions.

### 2.2. Preparation of Membranes

The sulfonation of montmorillonite (S-MMT) was carried out according to Al-Mashhadani et al. [1]. The casting solution method was used to prepare all membranes. PEBAX 1657 was dissolved in a mixture of absolute ethanol and water (7:3 *v*/*v*) at a concentration of 3 *w/v*% while stirring under reflux conditions for 2 h at 80 °C. Similarly, 10 *w/v*% PVA was dissolved in a mixture of water and absolute ethanol (7:3 *v*/*v*) under stirring and reflux for 3 h at 90 °C. After the complete dissolution of the PVA polymer and cooling of the solution, the SSA solution was added in an amount to obtain a membrane containing 40 wt% SSA. The resultant mixture was then stirred continuously at ambient temperature for 24 h. Varied proportions of S-MMT (3 wt%, 5 wt %, and 7 wt %) were then introduced into the mixed solutions of S-PVA and PEBAX 1657, maintaining an 8:2 blend ratio, and they were stirred continuously for 4 h at room temperature (25 °C). In the following, sulfonated polyvinyl alcohol PVA and poly(ether-block-amide) PEBAX1657 blend membranes are referred to as SPP. The prepared solutions were spread onto glass Petri dishes and left to dry for over 48 h at room temperature (25 °C). The procedure for membrane production is illustrated in Figure 1. The name of the blend membranes is shown in Table 1.

As a reference, a recast Nafion membrane was prepared following the procedure described in reference [10]. In short, we used a solution-casting method. First, the i-PrOH solvent of the 5 wt% Nafion solution was changed to DMAc; then, it was poured into glass Petri dishes, and the solvent was evaporated at 80 °C for 24 h. After this, the membranes were heat treated for 4 h in an oven at 120 °C. Finally, to fully protonate the recast Nafion, it was treated with 0.5 M H_2_SO_4_ at 80 °C for 1 h.

### 2.3. Membrane Characterization

For each sample, three parallel tests were performed using different characterization techniques, which enabled the calculation of the mean and standard deviation values.

#### 2.3.1. Water Uptake (WU) and Swelling Ratio (SR)

To evaluate the membranes’ water uptake (WU) and swelling ratio (SR), square specimens with dimensions of 2 × 2 cm^2^ were immersed in deionized (DI) water for 24 h. Following immersion, excess surface water was blotted off with tissue paper, and the weight and size of wet membranes were recorded. The samples were then dried overnight in a vacuum oven at 50 °C. After drying, their mass and dimensions were measured again. Water uptake and swelling ratio were calculated using Equations (1) and (2).
(1)$$\mathrm{WU}\;(\%)=\frac{\left(Wwet-Wdry\right)}{Wdry}\times 100$$
where W_wet_ and W_dry_ denote the weight of the wet and dry samples, respectively.
(2)$$\mathrm{SR}\;(\%)=\frac{\left(Awet-Adry\right)}{Adry}\times 100$$

The terms “Adry” and “Awet” refer to the dry and wet areas of the membrane.

#### 2.3.2. Ion Exchange Capacity (IEC)

The ion exchange capacity (IEC) of the developed membranes was measured by acid-base titration and is expressed in millimoles equivalent per gram (meq/g). To start, the samples were soaked in 20 mL of 1 M NaCl solution for 24 h to exchange H^+^ with Na^+^. Afterwards, the solution containing displaced H^+^ was titrated with a 0.1 M NaOH solution. The IEC was calculated based on Equation (3).
(3)$$\mathrm{IEC}\;(\mathrm{meq}/g)=\frac{\left(CNaOH\times VNaOH\right)}{Wdry}$$
where C_NaOH_, V_NaOH_, and W_dry_ represent the NaOH titration solution’s concentration, the volume of NaOH used, and the dry weight of the membranes, respectively. Before measuring the dry weight of the membrane samples, the membranes were dried in an oven at 60 °C for 24 h to ensure the removal of any residual moisture and solvents. This drying temperature was chosen to avoid changing the membrane structure while ensuring thorough drying.

#### 2.3.3. Thermogravimetric Analysis (TGA) and Thermogravimetry/Mass Spectrometry (TG/MS)

The thermal stability of the components and the membranes was analyzed by a TG-MS using a Perkin Elmer TGS-2 thermobalance (Eurotherm, UK) equipped with an upgraded furnace and a temperature controller based in. This thermobalance was linked to a Pfeiffer HiQuad quadrupole mass spectrometer. Approx. 1 mg (TGA) and approx. 3 mg (TG/MS) samples were carefully placed into a platinum sample holder and flushed with argon gas at a flow rate of 140 mL/min. The samples were heated at a rate of 20 °C/min from RT up to 900 °C. In the case of TG/MS analysis, the evolved gases were then directed into the mass spectrometer via a heated capillary line. The MS was operated using an ionization energy of 70 eV. The ion intensities that were measured were normalized to account for the intensity of the ^38^Ar isotope in the carrier gas and the initial amount of the sample.

Adjustments were made to the measured ion intensities to compensate for the ^38^Ar isotope’s intensity in the argon carrier gas and the initial sample mass.

#### 2.3.4. Scanning Electron Microscopy (SEM)

The surface texture of the composite membranes was examined by scanning electron microscopy (SEM) using a ZEISS EVO 40XVP, Carl Zeiss AG, based in Oberkochen, Germany, manufactures the ZEISS EVO 40 XVP equipment. which is a variable pressure scanning electron microscope (VP-SEM). This microscope is well suited for performing microstructural examinations under a high vacuum condition. The cross-section of the composite membranes was investigated using a JEOL JSM-6380 LA variable pressure scanning electron microscope (VP-SEM). JEOL Ltd., based in Tokyo, Japan, manufactures the JEOL JSM-6380 LA equipment.

#### 2.3.5. Fourier-Transform Infrared Spectroscopy (FT-IR)

This study of how S-MMT interacts with the composite membrane made from an S-PVA and PEBAX 1657 blend was carried out through Fourier-transform infrared spectroscopy (FT-IR) using the attenuated total reflectance (ATR) technique. This analysis was performed with a Varian 2000 (Scimitar Series) spectrometer (Varian Inc. Palo Alto, CA, USA), spanning a spectral range between 4000 and 600 cm^−1^. To acquire the FT-IR spectrum for each membrane, an average of 64 scans were taken at a spectral resolution of 4 cm^−1^.

#### 2.3.6. X-ray Diffraction (XRD)

The X-ray diffraction (XRD) patterns of the S-MMT nanofiller within the S-PVA: PEBAX 1657 blend membranes with different S-MMT nanofiller contents were recorded using a Philips PW 3710 goniometer with a PW1050 Bragg-Brentano Para focusing system from Malvern Panalytical Ltd., Prague, Czech Republic. The X-ray emissions were generated using Cu Kα radiation, having a wavelength (λ) of 0.15418 nm. The range of diffraction angles (2θ) spanned from 4° to 75°. Lattice parameters were calculated employing the Pawley fitting technique, which is based on the comprehensive fitting of the profile.

#### 2.3.7. Mechanical Stability

The mechanical characteristics of all membranes were evaluated with a universal testing machine (Zwick Z005 GmbH & Co., KG, Ulm, Germany). The membranes measuring 75 mm by 10 mm were subjected to mechanical testing at a velocity of 20 mm/min, starting from a grip separation of 35 mm.

#### 2.3.8. Chemical Stability

To evaluate the chemical resistance of the synthesized composite membranes, hot Fenton’s reagent was used, containing 3 wt% hydrogen peroxide (H_2_O_2_) and 4 ppm ferrous sulfate (Fe_2_SO_4_). Square samples with dimensions of 2 × 2 cm^2^ were prepared. These synthesized composite membranes were then oven dried at 60 °C for 24 h. Following this drying period, the samples were precisely weighed on an analytical balance. They were immersed in Fenton’s solution at 80 °C for another 24 h. The samples were then removed, dried, and reweighed to ascertain the weight loss percentage [40].

#### 2.3.9. Performance of Membrane Electrode Assemblies (MEAs)

A QuinTech C-40-PT catalyst was used, with 40 m/m% Pt content. As Gas Diffusion Layers (GDLs), H23C6-type carbon paper from Freudenberg FCCT SE & Co. KG was purchased. The catalyst ink consists of 30 mg of catalyst, 320 μL of Nafion solution (5 m/m% Quintech NS05), and 320 μL of a.r. 2-propanol (99.99 *v*/*v*%, Molar Chemicals Kft (Halásztelek, Hungary)). In order to obtain Gas Diffusion Electrodes (GDEs), catalyst ink was spray coated onto the GDLs with an AB200-type airbrush (Conrad Electronic SE, Berlin, Germany) [41].

Our homemade membranes were first protonated by soaking the membranes in a 0.5 M sulfuric acid solution for 1 h and then rinsing them with distilled water.

The MEAs were made by hot pressing the layers at 90 °C and 4.5 t pressure for 4 min, according to the following layer order: cathode GDE-tested membrane anode GDEs. The active area of MEAs is 16 cm^2^ with 0.15 mg Pt/cm^2^, and both are on the cathode and anode sides. Detailed information on the MEAs is listed in Table 2.

Bipolar plates (BPPs) consist of Ti (grade 1 purity). The channel has a single serpentine shape, and its cross-sectional dimensions are 1.2 × 1.2 mm^2^ (width × depth). The size of the GDE/active zone is 40 × 40 mm^2^, and the PEM (and the BPP) is 70 × 70 mm^2^.

Fuel cell tests were carried out on a Scribner 850 Fuel Cell Test System apparatus. MEAs were placed in a single cell within gaskets on both the anode and cathode sides, and 150 μm thick gaskets and MDPE foil (BRALEN FA 03-01) were purchased from Mizseplast Kft., Budapest, Hungary. The outer dimensions of the gaskets were 70 × 70 mm^2^, with a 41 × 41 mm^2^ window inside. Before the tests, the MEAs were activated in humidified H_2_ (saturated at 80 °C, 80% RH) and 200 mL/min O_2_ (saturated at 80 °C, 80% RH) on the anode and cathode sides, respectively, at 80 °C and 0.0625 A/cm^2^ load for 1 h. The polarization measurements were started with an open voltage; then, the current was increased in steps of 0.2 A with a duration of 120 s. Polarization was stopped when the voltage dropped below 80 mV.

## 3. Results and Discussion

### 3.1. Water Uptake Capacity and Swelling Ratios of Prepared Membranes

Figure 2 shows the water uptake and swelling ratios of the membranes. The water uptake increased with increasing sulfonated montmorillonite (S-MMT) content, while the swelling ratio decreased monotonically. The presence of the sulfonyl group in S-MMT could improve hydrophilicity [36]. This property increases the ability of the membrane to absorb water molecules. Additionally, the porous nature of MMT may improve capillary action, facilitating the draw and movement of water into its interstitial spaces. The spacing between MMT layers may accommodate a large amount of water molecules [35].

In general, in-plane swelling is a harmful phenomenon, as it leads to dimensional changes and, accordingly, stress between the layers of the membrane electrode assembly. Nevertheless, with increasing S-MMT content in the blend membrane, the swelling ratio was successfully mitigated. The excess water absorbed compared to the unfilled membrane seems to fit into the interstitial spaces of S-MMT, which ultimately does not lead to in-plane swelling of the membrane.

In conclusion, composite membranes containing 7% SPP S-MMT show promise for improving the performance of PEMFCs at low humidity. In terms of water uptake and swelling ratio, compared to recast Nafion, our composite membranes showed significantly better results [1,42].

### 3.2. Ion Exchange Capacity (IEC) of Prepared Membranes

The ion exchange capacity (IEC) of a membrane quantifies the number of cations that can be exchanged for protons, which is a vital metric for the suitability of membranes in fuel cell technologies. Figure 3 presents the IEC findings for the variously blended membranes, illustrating that an uptick in S-MMT concentration correlates with a rise in the IEC compared to the parent SPP membrane. Accordingly, the membrane containing 7% S-MMT showed the highest IEC at 0.46 meq/g, whereas the undoped SPP membrane with 0% S-MMT demonstrated an IEC of 0.31 meq/g. This enhancement in the IEC is ascribed to the sulfonic acid groups on the surface of sulfonated montmorillonite, which increase the concentration of ion exchange sites compared to the unfilled membrane [43]. We have previously demonstrated that membranes with a higher IEC exhibit increased ion mobility too, leading to higher ion conductivity [1]. Not surprisingly, the surface sulfonic acid groups of S-MMT play a crucial role in the enhanced water absorption capacity, increased IEC, and ionic conductivity. Therefore, it is logical to assume that the usage of the new SPP membrane with 7% S-MMT filler improves the performance of the membrane electrode assembly in a fuel cell. Nevertheless, compared to recast Nafion, composite membranes containing S-MMT show smaller IECs [42]. Although sulfonation provides ion exchange sites, the composite membrane still contains fewer sulfonic acid groups than Nafion. We plan to further modify the sulfonation conditions and design PEBAX-based versatile polymer blends in our laboratory to enhance the IEC.

### 3.3. Thermogravimetric Analysis (TGA) and Thermogravimetry/Mass Spectrometry (TG/MS)

Thermogravimetric analysis was used to study the thermal stability of the components (Figure 4A) and the new membranes (Figure 4B) as well.

Figure 4A(a) represents the TG, while Figure 4A(b) shows the DTG curves of the components. The thermal decomposition of S-MMT starts at around 200 °C and ends at around 600 °C. The mass loss of the sample could be due to the scission of the sulfonated groups. The mass loss of S-MMT is no more than 15% at 900 °C. PEBAX decomposes from 350 to 500 °C under an inert atmosphere. The decomposition range is narrower compared to that of S-MMT, and almost all the samples devolatilized up to 900 °C. On the TG and DTG curves of the S-PVA, a three-step thermal decomposition process can be seen as well on the TG and DTG curves of the SPP blend membranes (Figure 4B).

The TG curves of the SPP blend membranes in Figure 4B(a) are quite similar, so only the DTG and mass spectrometric curves of sample SPP 7% S-MMT are presented in Figure 4B(b). The first thermal decomposition step (by ~30% loss of all membranes) takes place between 50 and 230 °C and is attributed to the evaporation of the adsorbed water represented by the *m*/*z* 18 curve.

The second weight loss of the samples occurs because of the evolution of sulfur dioxide and carbon dioxide between 230 and 350 °C. Sulfur dioxide is represented by the evolution profiles of fragment ion *m*/*z* 48 (SO^+^) and molecule ion *m*/*z* 64 (SO_2_^+^). These ions are originated from the sulfonate functional groups. Carbon dioxide (*m*/*z* 44 ion) is a typical thermal decomposition product of the organic materials. Decomposition of PEBAX and the remaining part of the poly(vinyl alcohol) take place under the third mass loss step between 390 and 580 °C [44]. One of the most characteristic thermal decomposition products of the PVA is acetone (*m*/*z* 43 and 58), which can be seen in Figure 4B(b). There is no significant difference between the temperature range and magnitude of the different decomposition steps obtained on different membranes. Since the residual mass, summarized in Table 3, is the sum of the carbonaceous residue and the clay (MMT), we can say that the decomposition of the polymer is somewhat more pronounced in the presence of the filler. Nonetheless, the integration of the S-MMT filler does not significantly alter the thermal degradation pattern of the membranes.

### 3.4. Scanning Electron Microscopy (SEM) Analysis

The thermal, mechanical, and electrochemical properties of the membranes, and thus their performance in fuel cells, largely depend on the compatibility of S-MMT and polymers, as well as the miscibility of these components.

The surface morphology and the cross-section of the samples were analyzed by Scanning Electron Microscopy (SEM). SEM images presented in Figure 5 indicate the homogenous distribution of the S-MMT filler within the SPP matrix on all membranes, even at a content of 7%. Overall, the fabricated composite membranes display a cohesive, uniform, and smooth appearance. Uniform dispersion can be a prerequisite for better properties of the proton exchange membrane [45,46]. For example, a uniformly dispersed electrospun TiO_2_/ZrO_2_ nanofibrous web was supposed to lead to about two times higher water retention and thirty times less dimensional change than a pure Aquivion membrane under in-water membrane hydration conditions [47].

Recast Nafion nanocomposite membranes were loaded with ZrO_2_ nanoparticles (R-Nafion/ZrO_2_). The generated nanocomposite membranes are reported to suit PEMFC applications because the zirconia nanoparticles produced a dense microstructure and excellent nanoparticle dispersion in the membrane matrix [48].

Carbon nanotubes coated with silica (SCNTs) were synthesized utilizing the simple sol–gel technique to create chitosan/SCNT (CS/SCNT) composite membranes. The silica layer of the CNTs enhanced the interaction of the SCNTs with the chitosan, resulting in uniform SCNT dispersion in the composite membrane. According to this study, composite membranes made of CS and SCNTs have a bright future as proton exchange membranes [49].

### 3.5. Fourier-Transform Infrared Spectroscopy (FTIR) Analysis

Fourier-transform infrared (FT-IR) spectroscopy was used to identify the functional groups and investigate the interactions between the SPP membranes and the S-MMT filler. Spectra are shown in Figure 6.

In general, the spectra of the SPP membranes shown in Figure 6A consist of bands attributable to the functional groups found in the various constituents of the membranes (shown in Figure 6B). There are stretching vibrations of C-O-C groups at 1020 cm^−1^, C-C bonds at 1130 cm^−1^, and carboxylate groups at 1416 cm^−1^. The peaks observed at 2940 cm^−1^ and 3150 cm^−1^ can be attributed to the stretching vibrations of the (C-H) and (O-H) groups, respectively. An absorption band assigned to stretching vibration of the ester carbonyl (-COO-) groups was observed around 1730–1735 cm^−1^, indicating the presence of SSA-modified PVA, i.e., the esterification of carboxyl groups in SSA with hydroxyl groups in PVA [39,50]. As can be seen in Figure 6B, modification of PVA with SSA leads to the appearance of this new absorption band.

However, the new membranes are not simple physical mixtures of different components; interactions can be discovered between the constituents.

A hydrogen bond interaction between the hydroxyl (OH) groups in PVA and the carbonyl (C=O) groups in the amide segments of PEBAX can be assumed, which is manifested in the red shift of the absorption band assigned to the stretching vibration of the carbonyl group at 1630 cm^−1^ when compared to the spectrum of pure PEBAX 1657, where the corresponding band was detected at 1650 cm^−1^ [1]. This spectral feature strongly overlaps with that of SSA carboxylate, so we only see a broadening of the absorption band at 1730 cm^−1^ in the membrane blends. This interaction strongly contributes to enhancing the compatibility and adhesion of interfaces between PVA and PEBAX in blends.

Figure 6A also illustrates the slight change in spectra upon the addition of sulfonated montmorillonite (S-MMT). It is assumed that a hydrogen bond interaction may occur between the hydroxyl groups of sulfonic acid groups in MMT and the amide groups of PEBAX 1657. Accordingly, the intensity ratio of peaks at 1630 cm^−1^ to that at 1730 cm^−1^ increases with increasing S-MMT content. At low S-MMT concentration, there is only one shoulder at 1630 cm^−1^, while in the presence of 7% S-MMT, a distinct peak appears at this position. Table 4 summarizes the key absorption bands, their assigned functional groups, and the observed changes due to various interactions within the SPP membranes.

### 3.6. X-ray Diffraction (XRD) Analysis

X-ray diffraction was applied to identify how crystallinity is affected in various SPP membranes containing different amounts of S-MMT filler. Polymer components of the blend membranes are semicrystalline in nature. PEBAX 1657 is a copolymer that consists of both crystalline polyamide (PA) and amorphous polyethylene oxide (PEO) segments.

PEBAX 1657 generally shows characteristic crystalline peaks due to the presence of polyamide segments, with the dominant peak around 24° [33,51]. The X-ray pattern highlights the semicrystalline nature of the polymer, with the amorphous polyether segments contributing to the background [33,51].

As emerges in Figure 7, four reflections are observed at 2θ angles of approximately 14.7°, 17.1°, 20°, and 25°. The peaks observed at 14.7° and 17.1° indicate the presence of crystalline PA segments [52]. Furthermore, PVA also has a partially crystalline structure. As described in reference [53], the X-ray pattern of PVA shows a broad peak, indicating its semi-crystalline nature. The [54] intermolecular hydrogen bonds formed between the PVA chains lead to the peak around 2θ = 19.3° [22,53]. After sulfonation with SSA (S-PVA), the presence of SSA in PVA reduces the number of intermolecular hydrogen bonds. The decrease in crystallinity results in a broader and less intense peak, indicating the breakdown of the PVA crystal structure due to the introduction of sulfone groups [54]. Although the presence of SSA in PVA reduces the number of intermolecular hydrogen bonds, a higher amount of SSA results in a new crystalline phase with a reflection at around 2θ≈25°. Even though SSA-modified PVA dominates the blends in terms of weight percentage, the crystallinity of this component can hardly be detected and can only be clearly evidenced when the percentage of S-MMT filler is above 3%. The results indicate that the membranes became more crystalline as the amount of S-MMT filler increased. It should be mentioned that MMT is a crystalline clay, and the X-ray pattern of MMT shows a peak around 6°, corresponding to the basal spacing of the clay layers [38]. After sulfonation (S-MMT), this peak may shift or lose intensity because the addition of sulfone groups changes the interlayer spacing. The low intensity of the diffraction peaks can also indicate the exfoliation of the clay [38]. As a result, the presence of only 7% S-MMT cannot be detected by XRD.

Nevertheless, the good compatibility of the S-MMT nanofiller and the SPP blend membranes gradually improves the crystallinity of our blend membranes, reaching its highest degree at 7% filler content, as shown in Figure 7. Our hypothesis is that colloidal S-MMT particles behave as nucleation centers facilitating the nucleation of polymer crystallites. Accordingly, although 7% S-MMT cannot contribute to the X-ray pattern of the blends, it is still present in a large enough amount to initiate the crystallization of the polymer blend.

### 3.7. Mechanical Properties

The mechanical strength of membranes is a key characteristic. In general, the addition of inorganic fillers to hybrid membranes has proven to be an effective and simple method to improve the mechanical properties of polymers [45,52]. Tensile tests were performed to evaluate the effect of S-MMT nanofiller on the mechanical properties of SPP membranes. The results obtained are shown in Figure 8.

The membrane blend without S-MMT exhibited the lowest performance at a tensile stress of 3.4 MPa, while increasing the amount of S-MMT nanofiller improved the mechanical properties of membranes. Tensile strengths of 4.4, 5.3, and 6.5 MPa were obtained on SPP membranes containing 3%, 5%, and 7% S-MMT, with notable elongations at the break. This improvement is likely due to the enhanced interfacial interaction between S-MMT, S-PVA, and PEBAX 1657. Hydrogen bonds between the polymer chains, as well as electrostatic interactions with S-MMT, increase interfacial adhesion. Additionally, higher crystallinity of materials with S-MMT also contributes to improved mechanical properties. In summary, increasing the amount of S-MMT filler improved the compatibility and miscibility of different components in the membrane blends, improved the long-range order and hence the cohesion within crystalline phases, which ultimately facilitates load transfer under tension. This improves the mechanical properties of the composite. Similar to other characteristics investigated above, the composite membrane containing 7% S-MMT showed the best properties. The properties improve monotonically with increasing S-MMT content. We have not reached the optimum yet.

It should also be emphasized that in terms of tensile strength, the mechanical properties of Nafion membranes are better than those of our new blends. In our earlier study, we obtained 19.9 MPa and close to 24 MPa tensile strength for a recast Nafion and Nafion XL, respectively. Nevertheless, elongation at break values is excellent for SPP blend membranes. In the case of Nafion XL, only 60% elongation was obtained, while in the case of SPP mixtures, as shown in Figure 8, we obtained higher values by an order of magnitude [42].

### 3.8. Chemical Stability Test

Fenton’s reagent test, a common method for assessing chemical stability, is employed to determine the resilience of proton exchange membranes (PEMs) against radical species (OH·and OOH) generated at the electrodes in a PEM fuel cell.

Figure 9 shows the weight loss of the prepared membranes as a function of S-MMT content after exposure to Fenton’s reagent test; as the filler content increased, the weight loss decreased. This might be because S-MMT has antioxidant or free radical-scavenging properties [55]. Accordingly, the presence of S-MMT in the blend has the potential to mitigate the effects of hydroxyl radicals generated during the Fenton reaction, and in general, it is expected that S-MMT can reduce oxidative damage to the polymer matrix [45]. According to the related art, the membrane blends were more stable in oxidative environments when amide groups from the PEBAX 1657 chain were added [56]. Eventually, the membrane blends containing PEBAX 1657 and S-MMT succeeded in preserving their structural cohesion, showing no punctures or fractures. We also observed a monotonous improvement in chemical stability with increasing S-MMT filler content, and there is still room for improvement. However, the synthesized membranes already showed better chemical stability than the recast Nafion, which showed a 6% weight loss in the Fenton test [1].

### 3.9. Fuel Cell Performance Test

The electrochemical performance of membrane electrode assemblies (MEAs) was evaluated in a single fuel cell at 80 °C and 80% relative humidity for both H_2_ and O_2_. Polarization and power density curves are presented in Figure 10. The voltage current density polarization curves run one above the other with increasing filler content, i.e., the voltage increases at a given current density value. Moreover, as the S-MMT content increases, there is a remarkable increase in the peak power density values. The current and power density values of the SPP membrane with 7% S-MMT content are 149.5 mA/cm^2^ and 49.51 mW/cm^2^, respectively.

In our strategy for the development of the new fluorine-free membrane, we took into account several possible factors determining the final electrochemical properties, such as a good balance between hydrophobic and hydrophilic polymer chains as well as the crystalline and amorphous parts and the optimal ratio of sulfonated components. All these factors strongly influence characteristics that are not actually independent of each other, such as ion exchange capacity, swelling ratio and proton conductivity, and mechanical and chemical stability. The polymer blend containing the SSA-modified PVA and PEBAX copolymer and sulfonated MMT works well and, according to our hypotheses, with further modification, it still can be improved. The current performance improvement with increasing filler content is in good accordance with the ion exchange capacity (IEC) and water uptake results.

However, there is a lot to do. As a reference, recast Nafion treated appropriately in sulfuric acid was applied under the same conditions in the single-cell test. As can be seen in Figure 10, this membrane still outperforms the membranes designed in our laboratory.

Nevertheless, the observed monotonic improvement in performance with increasing filler content in our membranes encourages us to continue this track by systematically changing the polymers, sulfonating them, and adding different sulfonated inorganic fillers.

## 4. Conclusions

Innovative composite membranes were designed and prepared, consisting of S-PVA and PEBAX with a mixing ratio of 8:2, and explored. These membranes were produced by introducing different proportions (0%, 3%, 5%, and 7%) of sulfonated montmorillonite (S-MMT) filler using a solution casting method for fabrication. The addition of higher amounts of S-MMT has resulted in notable improvements in several key properties of the SPP blend membranes, including enhanced ion exchange capacity (IEC), superior mechanical and thermal stability, and increased water uptake. Furthermore, the loading of S-MMT into the SPP blend membranes positively affected their performance in fuel cell studies. The membrane labeled SPP 7% S-MMT demonstrated the highest current and power density with values of 149.5 mA/cm^2^ and 49.51 mW/cm^2^, respectively. The performance of the new membranes has significant potential for use in proton exchange membrane fuel cell (PEMFC) technologies. Although the single-cell test results are below those of the recast Nafion membrane, these outcomes encourage further refinements in membrane design. In further expanding the experimental space, we insist that these membranes are made entirely from non-fluorinated substances and may contain inorganic materials as a filler.

## Figures and Tables

**Figure 1 membranes-14-00211-f001:**
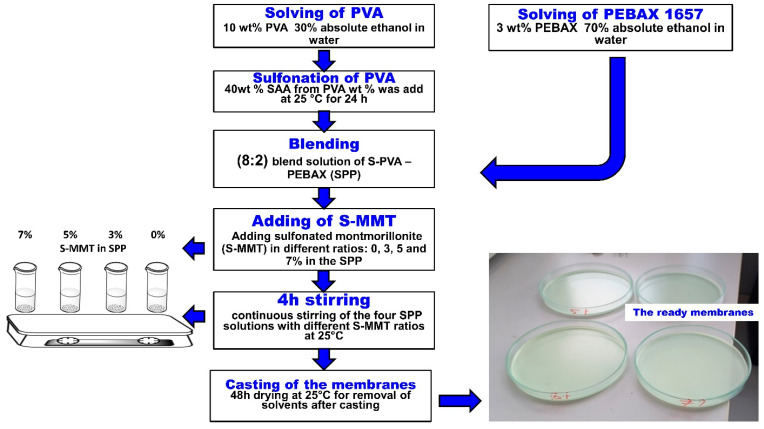
Manufacturing procedure of blend membranes with different S-MMT content.

**Figure 2 membranes-14-00211-f002:**
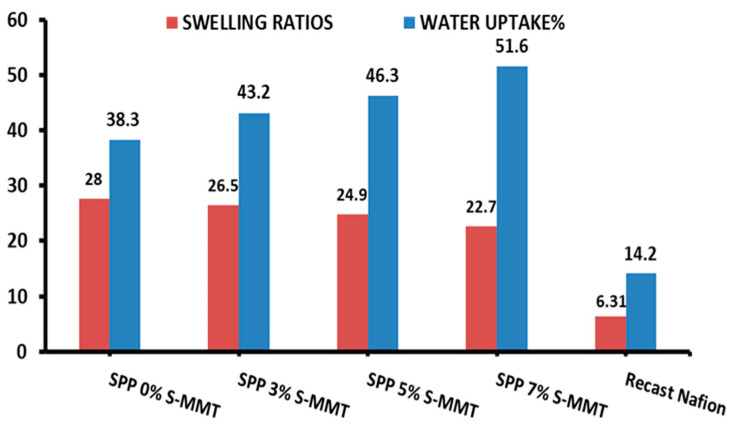
Water uptake and swelling ratio of SPP blend membranes and recast Nafion.

**Figure 3 membranes-14-00211-f003:**
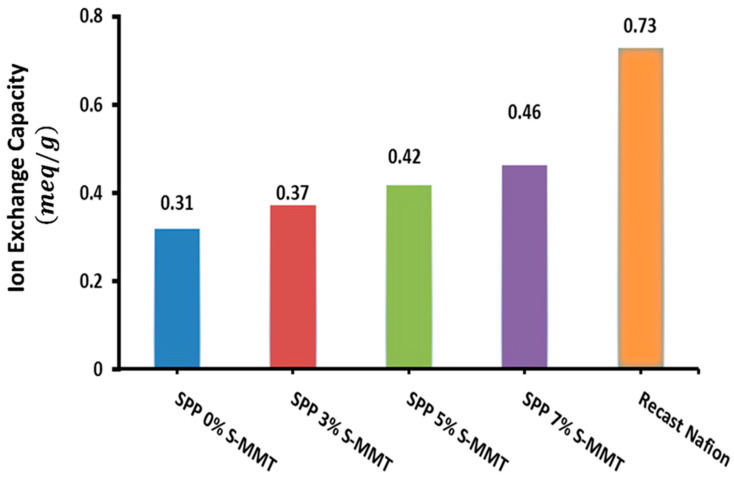
Ion exchange capacity for SPP blend membranes and recast Nafion.

**Figure 4 membranes-14-00211-f004:**
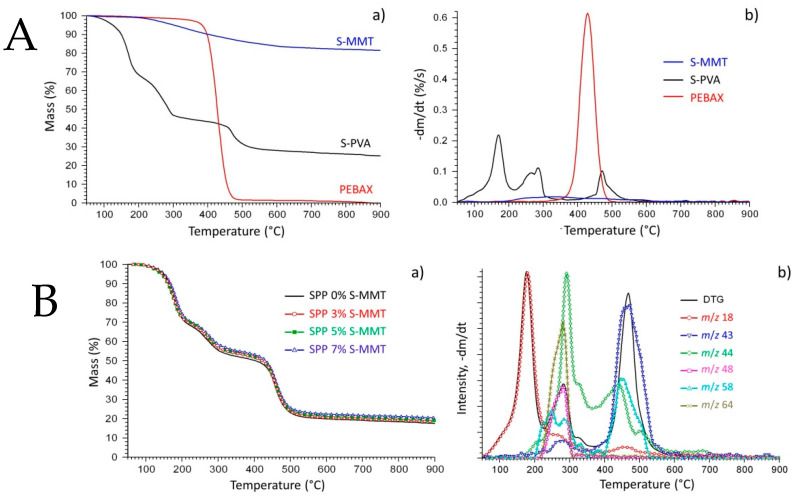
(**A**): (**a**) Thermogravimetric (TG) and (**b**) derivate thermogravimetric (DTG) curves of S-PVA, PEBAX, and S-MMT; (**B**): (**a**) TG curves of the SPP blends membranes and (**b**) DTG curve and evolution profiles of some characteristic mass spectrometric ions of the SPP 7% S-MMT sample, *m*/*z*: 18 (water), *m*/*z*: 43 and 58 (acetone), *m*/*z*: 44 (carbon dioxide), *m*/*z*: 48 and 64 (sulfur dioxide).

**Figure 5 membranes-14-00211-f005:**
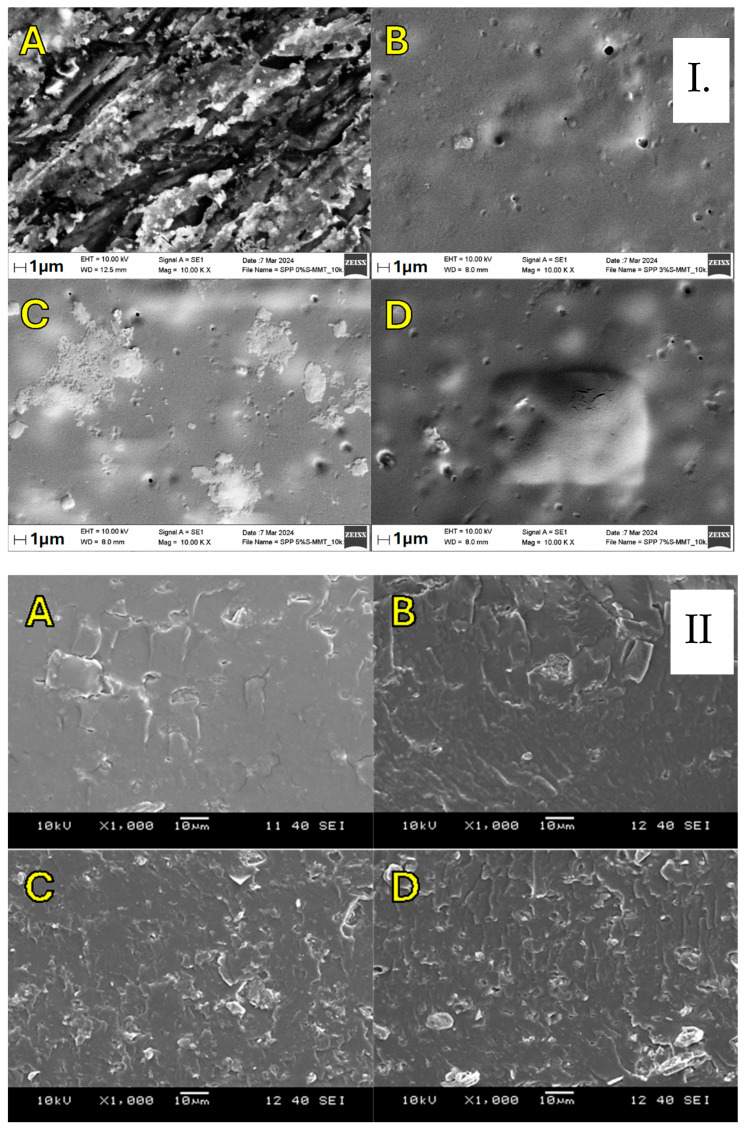
In-plane and cross-sectional SEM images (**I** and **II**) for blend membranes ((**A**–**D**) refer to SPP 0% S-MMT, SPP 3% S-MMT, SPP 5% S-MMT, and SPP 7% S-MMT, respectively).

**Figure 6 membranes-14-00211-f006:**
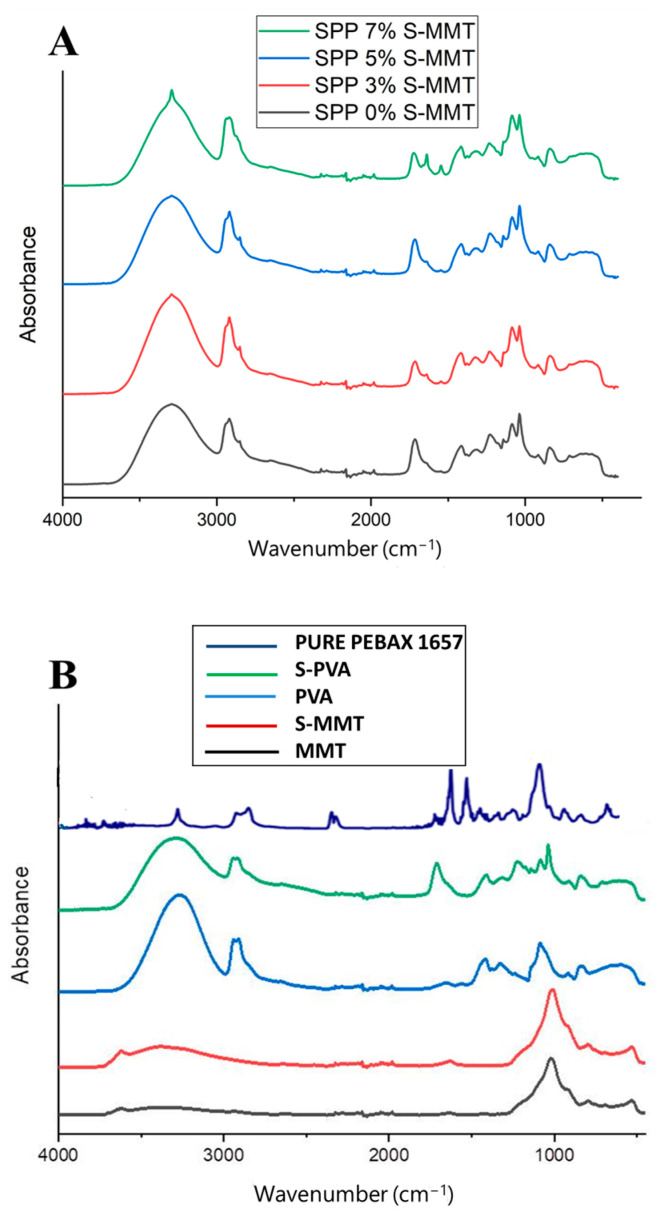
FTIR spectra of (**A**) SPP blend membranes and (**B**) the different constituents, such as parent and sulfonated montmorillonite (MMT and S-MMT), PVA, and S-PVA, as well as the pure PEBAX 1657.

**Figure 7 membranes-14-00211-f007:**
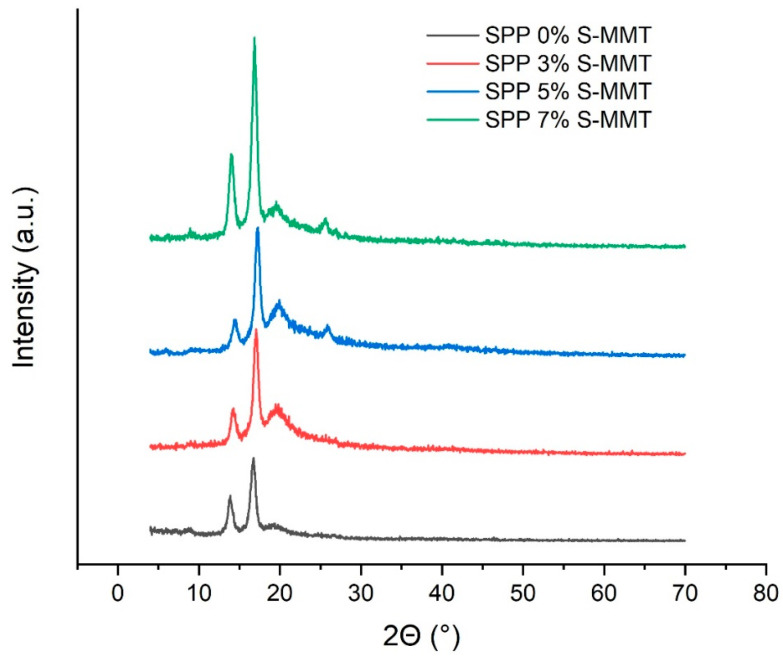
X-ray diffraction patterns of SPP blend membranes.

**Figure 8 membranes-14-00211-f008:**
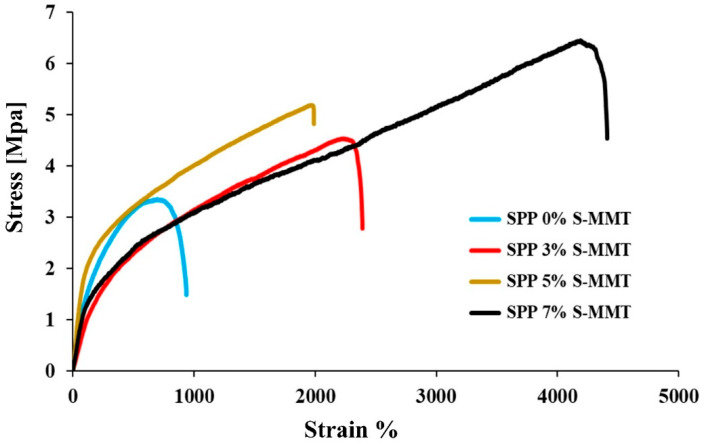
Mechanical stability for SPP blend membranes.

**Figure 9 membranes-14-00211-f009:**
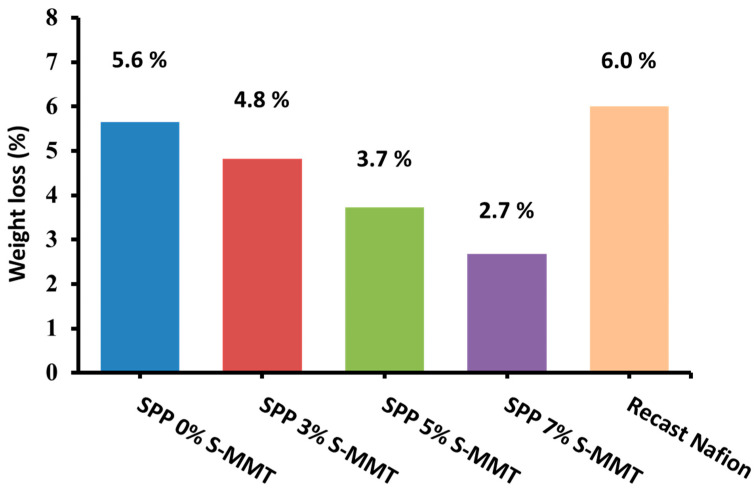
Chemical stability by Fenton’s test for SPP blend membranes and recast Nafion.

**Figure 10 membranes-14-00211-f010:**
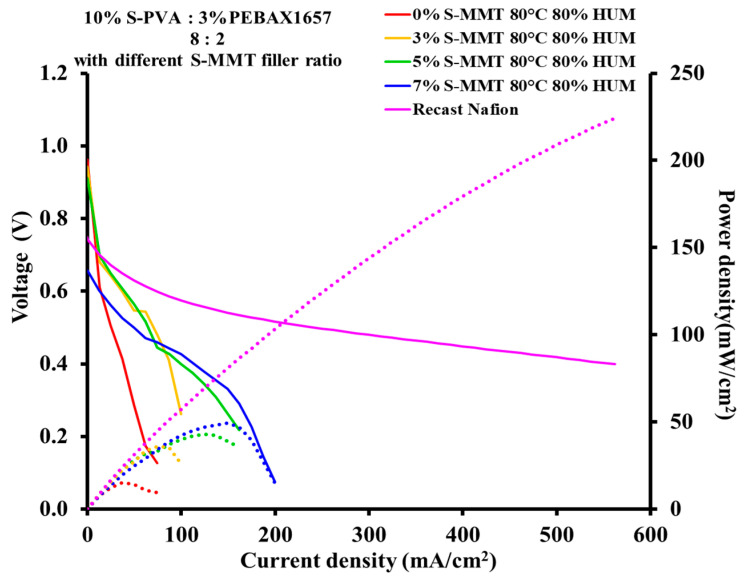
Polarization curves and power density curves for SPP blend membranes. Solid and dashed lines correspond to voltage and power density.

**Table 1 membranes-14-00211-t001:** Name of the membranes loaded with different amounts of S-MMT.

Composition of Membrane	Name of Membrane
S-PVA 8:2 PEBAX1657 without S-MMT	SPP 0% S-MMT
S-PVA 8:2 PEBAX1657 with 3% S-MMT	SPP 3% S-MMT
S-PVA 8:2 PEBAX1657 with 5% S-MMT	SPP 5% S-MMT
S-PVA 8:2 PEBAX1657 with 7% S-MMT	SPP 7% S-MMT

**Table 2 membranes-14-00211-t002:** Detailed information of the MEA.

Component	Description	Type
GDL	40 × 40 × 0.250 mm^3^	H23C6-type carbon paper from Freudenberg FCCT SE & Co
Catalyst	Loading: 0.15 mg Pt/cm^2^, both on the cathode and anode sides	QuinTech C-40-PT with 40 m/m% Pt content
Catalyst ink	30 mg of catalyst, 320 μL of Nafion solution, and 320 μL of 2-propanol	Nafion solution (5 m/m% Quintech NS05) 2-propanol (99.99 *v*/*v*%, Molar Chemicals Kft.)
PEM	70 × 70 mm^2^. Our homemade membranes were soaked with 0.5 M sulphuric acid	
Gasket	70 × 70 × 0.15 with a 41 × 41 mm^2^ window	BRALEN FA 03-01

**Table 3 membranes-14-00211-t003:** Residue (%) obtained after the TGA of the membranes.

Sample	Residue (%)
SPP 0% S-MMT	17.36%
SPP 3% S-MMT	18.42%
SPP 5% S-MMT	19.01%
SPP 7% S-MMT	20.45%

**Table 4 membranes-14-00211-t004:** Spectral features and corresponding wavelengths in SPP membranes.

Wavelength (cm^−1^)	Functional Group/Bond	Observation/Description
1020 cm^−1^	C-O-C stretching vibration	Attributable to ether groups in the membrane components [32,39,46].
1130 cm^−1^	C-C bond	Stretching vibration in the membrane constituents [32,39,46].
1416 cm^−1^	Carboxylate groups	Stretching vibration indicates the presence of carboxylate in the membrane [32,39,46].
1630 cm^−1^	Carbonyl (C=O) in SPP S-MMT blends	Red shift due to hydrogen bonding between OH groups in PVA and C=O groups in PEBAX. Peak intensity increases with S-MMT content.
1650 cm^−1^	Carbonyl (C=O) stretching in pure PEBAX	Indicates a minor shift when compared to the spectrum of PEBAX with hydrogen bonding interactions [1].
1657 cm^−1^	Amide groups of PEBAX	Reference for pure PEBAX. Overlaps in the presence of S-MMT, suggesting interaction with hydroxyl groups in sulfonic acid [1].
1730–1735 cm^−1^	Ester carbonyl (-COO-) groups	Appearance due to esterification of carboxyl groups in SSA with hydroxyl groups in PVA, indicating the presence of SSA-modified PVA [32,39,46].
2940 cm^−1^	C-H stretching vibration	Attributed to stretching vibrations of the C-H groups present in the membrane components [32,39,46].
3150 cm^−1^	O-H stretching vibration	Represents stretching vibrations of O-H groups, which is indicative of hydroxyl content in the membranes [32,39,46].

## Data Availability

The data presented in this study are available on request from the corresponding author. The data are not publicly available due to their unstandardizable complexity.

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
