# Peer review of "Novel, Fluorine-Free Membranes Based on Sulfonated Polyvinyl Alcohol and Poly(ether-block-amide) with Sulfonated Montmorillonite Nanofiller for PEMFC Applications"

_membranes, 2024, doi:10.3390/membranes14100211_

Round 1

Reviewer 1 Report

Comments and Suggestions for Authors

In this work, novel blend membranes were prepared using a solution-casting technique. The effect of different amounts of S-MMT on water uptake, swelling degree, thermal stability, chemical stability, ion exchange capacity, and performance of the membranes are investigated.

Comments on the Quality of English Language

Minor editing of English language required.

Author Response

Thank you very much for taking the time to review this manuscript.

Reviewer 2 Report

Comments and Suggestions for Authors

1.The reviewer is curious if the authors have tried to continue to increase the proportion of S-MMT?If the proportion of S-MMT continues to increase, will the IEC, water uptake and swelling ratio, etc. of the composite film remain monotonically changed?

2.In Figure 8, why is the strain percentage of SPP 5%-s-MMT lower than that of SPP 3%-s-MMT, rather than showing a monotonous change.

3.In Figure 10, what causes the polarization curve of the SPP blend membranes not to be smooth

Author Response

Thank you very much for taking the time to review this manuscript. You will find our detailed answers attached in a separate file.

Reviewer 3 Report

Comments and Suggestions for Authors

The authors describe an interesting approach for the preparation of fluorine-free proton exchange membranes based on a blend of sulfonated polyvinyl alcohol and poly(ether-block-amide), with the addition of sulfonated montmorillonite as a nanofiller to improve chemical stability, heat resistance, and tensile strength. The manuscript is well-structured and provides an adequate spectrum of characterization techniques applied to the proposed membranes. However, the discussion of the results is generally poor and lacking depth. Thus, the following concerns should be addressed before recommending publication:

1. In the Introduction, the authors should expand the discussion on the use of PVA, PEBAX, and MMT in PEM fuel cells by presenting and commenting some more examples from the recent literature.

2a. In Section 2.2, the authors state that “The prepared solutions were spread onto glass Petri dishes and left to dry for over 48 h”. I assume that drying was carried out at room temperature, but it should be specified for the sake of clarity.

2b. Did the authors try to perform drying in an oven at higher temperatures to reduce the drying time? Are there any possible side effects in this regard?

3. The manufacturing procedure schematized in Figure 1 is a bit confusing in its structure and contains some sentence structure errors. It could be improved by reducing the written parts and by introducing numbers/letters to identify the different production steps.

4. In Section 2.3.2, the authors should state the drying conditions adopted before measuring the dry weight of the membrane samples used in ion exchange capacity tests.

5. The results of water uptake, swelling ratio, and ion exchange capacity should be compared with those of reference Nafion membranes and with those of similar novel membranes discussed in the literature. Moreover, the separate results of S-PVA and PEBAX should be discussed as well, for the sake of comparison with those of the blends.

6a. The discussion of thermogravimetric results does not seem to match the presented plots. The authors mention a 5% weight loss around 175 °C, attributed to the evaporation of moisture. Looking at Figure 4, the corresponding thermal phenomenon accounts for a 30% weight loss (and not 5%) in all membranes, which is unlikely to correspond to water evaporation only. Moreover, the authors state that “Decomposition of polymer carbon backbone and the membrane structure take place above 280 °C”. However, the slope of the thermograms changes significantly only above 400 °C, indicating that the decomposition takes place in the 400-500 °C range. These attributions should be carefully revised and supported with references.

6b. The thermograms of S-PVA, PEBAX, and S-MMT should be displayed and discussed as well, for the sake of completeness.

6c. In Section 2.3.3, the authors mentioned the use of a mass spectrometer to analyze the evolved gases. If present, the results of this analysis should be discussed as well together with thermogravimetric plots, in order to improve the discussion. Otherwise, the reference to mass spectrometry analyses should be removed from the manuscript.

6d. As the authors correctly stated, “There is no significant difference between the temperature range and magnitude of the different decomposition steps obtained on different membranes” and “the integration of S-MMT filler does not significantly alter the thermal degradation pattern of the membranes”. Therefore, writing in the abstract that the introduction of the filler “ultimately resulted in increased heat resistance” is somewhat misleading and it should be rephrased in a more appropriate way.

7a. Cross-sectional SEM images should be provided as well to demonstrate the good miscibility and compatibility of S-MMT and polymers, as an analysis limited to the surface is not sufficient.

7b. The sentence “This property gives hope that our membranes will show good properties in fuel cell tests” is too speculative and should be demonstrated with literature references or removed from the manuscript.

8a. Concerning the FTIR spectra shown in Figure 6, the sample SPP 7% S-MMT seems to be richer in PEBAX than the other ones, as suggested by the higher relative intensity of the corresponding bands (for instance, the sharp peak at about 3250 cm-1, which is more visible than in the other specimens). Could this apparent compositional difference be due to local inhomogeneities in the analyzed samples? If this is the case, the authors should acquire FTIR spectra on multiple areas of the surface of the samples, in order to verify the uniformity of the composition.

8b. The FTIR spectra of MMT and S-MMT do not show evident differences. Is there evidence, in these spectra, of the effective sulfonation of MMT?

9a. The diffractograms of PVA, S-PVA, PEBAX, MMT, and S-MMT should be displayed and discussed as well, for the sake of completeness.

9b. Moreover, the comment “It should be mentioned that although MMT is a crystalline clay, its amount is too small to be detected by XRD” seems to be in contradiction with the previous statement “The results indicate that the membranes became more crystalline as the amount of S-MMT filler increased”. If the amount of S-MMT is high enough to alter the membrane crystallinity, as hypothesized by the authors, it is also likely to be high enough to be detected in the diffractograms. The authors should clarify this point.

10. The mechanical behavior and chemical stability of the blends should be compared with those of reference Nafion membranes and with those of similar novel membranes discussed in the literature.

11. Throughout the manuscript, the blend membrane is sometimes referred to as SPP and sometimes as SSP. The authors should check and revise this inconsistency in the nomenclature.

Comments on the Quality of English Language

Minor typos and mistakes should be checked and fixed throughout the manuscript, as well as some non-English words, such as Poli(tetrafluoretilén) in the Introduction.

Author Response

(The authors gave the same response as above.)

Round 2

Reviewer 1 Report

Comments and Suggestions for Authors

The following comments can be considered.

1.The introduction section can be extended to describe the importance and novelty of this study.

2.The proportions of S-MMT (3 wt%, 5 wt %, and 7 wt %) were used in the experiments. Why?

3.A table can be added to give the detailed information of the MEA.

4.The descriptions of the flow field of the BPs can also be added.

5.The fuel cell was operated under the operating condition of 353.15K and RH80. The effect of RH can be investigated.

6.As reported in this study, the maximum power density of the novel membrane is lower than that of the Nafion membrane. It still has a big gap from practical application.

Comments on the Quality of English Language

Minor editing of English language required.

Author Response

(The authors gave the same response as above.)

Reviewer 3 Report

Comments and Suggestions for Authors

I believe the authors thoroughly addressed the issues raised during the first round of revision and significantly improved the quality of the manuscript, which is now worthy of publication in Membranes.

Comments on the Quality of English Language

A few mistakes in sentence construction and grammar still persist but they can be addressed in the proof-reading phase

Author Response

(The authors gave the same response as above.)

Round 3

Reviewer 1 Report

Comments and Suggestions for Authors

Some content in the Response to Reviewer 1 Comments is not in the revised manuscript.